# Mechanistic PDE Networks for Discovery of Governing Equations

**Adeel Pervez** [1]  **Efstratios Gavves** [2]  **Francesco Locatello** [1]

## Abstract

We present Mechanistic PDE Networks – a model for discovery of governing *partial differential equations* from data. Mechanistic PDE Networks represent spatiotemporal data as space-time dependent *linear* partial differential equations in neural network hidden representations. The represented PDEs are then solved and decoded for specific tasks. The learned PDE representations naturally express the spatiotemporal dynamics in data in neural network hidden space, enabling increased power for dynamical modeling. Solving the PDE representations in a compute and memory-efficient way, however, is a significant challenge. We develop a native, GPU-capable, parallel, sparse, and differentiable multigrid solver specialized for linear partial differential equations that acts as a module in Mechanistic PDE Networks. Leveraging the PDE solver, we propose a discovery architecture that can discover nonlinear PDEs in complex settings while also being robust to noise. We validate PDE discovery on a number of PDEs, including reaction-diffusion and Navier-Stokes equations. Source code will be made available at: https://github.com/alpz/mech-nn-discovery-pde

## 1. Introduction

Partial differential equations (PDEs) enjoy widespread use as interpretable, analytical models of spatio-temporal dynamics in all areas of science (Temam, 2024; Turing, 1952). The *development* of PDE models still, however, is largely a manual task that is performed by domain experts examining experimental data. The modern explosion of experimental and simulation data has underscored the need for data-driven machine learning models for discovery of governing equa-

tions (Brunton et al., 2020). In spite of progress (Brunton et al., 2016; Raissi et al., 2019; d'Ascoli et al., 2023), the development of machine learning discovery models that are flexible, handle complex nonlinear data and account for missing physics, while being robust to noise and missing data, is very much an open problem.

A recent discovery architecture for *ordinary* differential equations (ODEs) was demonstrated by Mechanistic Neural Networks (MechNN) (Pervez et al., 2024), enabling a flexible and expressive network design for modeling explicit, interpretable nonlinear ODEs. MechNNs embed a specialized, parallel ODE solver in the network, allowing explicit differential equation representations in neural networks, which can be employed for ODE discovery and prediction, both relevant in applications including general time series and numerical simulations (Pervez et al., 2024) and in climate science (Chen et al., 2025; Yao et al., 2024).

In this paper we focus on *governing equation discovery* and demonstrate Mechanistic Neural Networks for discovery of *partial differential equations* from data. We develop a PDE solver, NeuRLP-PDE, for parallel native differentiable solution of linear partial differential equations over multiple spatial dimensions (in addition to the time dimension). In contrast to ODEs, scaling is significantly more difficult in PDE solving, with memory requirements increasing exponentially with dimension and quickly becoming excessive even for mid-sized grids. With NeuRLP-PDE, *sparse computation* in all phases of the PDE solver ensures that the memory requirement of the solver over batches of PDEs remains limited for sizeable grids. We build the solver using constrained optimization and a sparse *multigrid* preconditioned iterative linear solver (Saad, 2003) for solving the large and sparse linear systems that appear in the constrained optimization. The NeuRLP-PDE solver is then incorporated into the PDE discovery architecture enabling the learning of governing PDEs directly from data.

The MechNN-PDE *discovery* architecture couples the NeuRLP-PDE solver with deep neural networks that parameterize partial differential equations. Two natural constraints on discovered differential equations are that they should be 1) *simple*: i.e., built from elementary functions and 2) *concise*: containing as few terms as possible. The MechNN-PDE architecture enables simplicity and concise-

---

[1]Institute of Science and Technology, Klosterneuburg, Austria [2]Informatics Institute, University of Amsterdam, Amsterdam, The Netherlands. Correspondence to: Adeel Pervez <Adeel.Pervez@ist.ac.at>.

*Proceedings of the 42nd International Conference on Machine Learning*, Vancouver, Canada. PMLR 267, 2025. Copyright 2025 by the author(s).

ness by building PDE expressions using a family of elementary basis functions, each with a learnable parameter. Since elementary functions are restrictive, for expressive modeling the basis functions inputs are parameterized by neural networks. The built PDE is then solved by NeuRLP-PDE to obtain the solution $u$. Loss terms ensure consistency so that the parameterized basis functions equal the true basis functions at convergence. A sparsity loss ensures that the discovered expressions are concise and the entire pipeline is optimized by gradient descent.

The proposed synthesis of neural networks and PDEs for discovery has the following advantages.

- Derivatives never have to be directly on data and are only computed internally by the solver.

- PDEs can contain arbitrary differentiable functions, going beyond generalized linear functions.

- Numerical solving together with neural networks provide robustness to noise and missing data.

- Memory-efficient batch-parallel PDE solving on GPU with a sparse multigrid-preconditioned iterative solver.

## 2. Mechanistic PDE Networks

A Mechanistic PDE Network consists of a *mechanistic encoder*, $f_{Enc}$, a *differentiable solver*, NeuRLP-PDE, specialized for linear PDEs, and an optional *decoder*, $f_{Dec}$. The input to the encoder are spatio-temporal data over time and one or more spatial dimensions, denoted $u_{\text{data}}(t, x_1, x_2, \ldots, x_{n_d})$. Given input $u_{\text{data}}$ the mechanistic encoder produces a set, of predefined size $n_{eq}$, of linear partial differential equations which we denote as $\{\mathcal{P}_{u_i}\}_{i \in [n_{eq}]}$ (using the notation $[n] := \{1, \ldots, n\}$). Each $\mathcal{P}_{u_i}$ is a linear PDE for a spatio-temporal function $u_i(t, x_1, x_2, \ldots)$.

All functions $u_i$ are defined over a domain $\mathcal{D}$ which we assume to be a Cartesian grid, i.e., $\mathcal{D} = \mathcal{T} \times \mathcal{X}_1 \times \mathcal{X}_2 \ldots \mathcal{X}_{n_d}$, where $\mathcal{T}$, $\mathcal{X}_i$ are intervals. For conciseness we collect the partial derivative indices appearing in PDE $\mathcal{P}_{u_i}$ into a multi-index set denoted $\mathcal{M}$. As an example $\mathcal{M} = \{t, x_1, x_2, x_1 x_2\}$, $\{u_m\}_{m \in \mathcal{M}}$ denotes the set of partial derivatives $\{u_t, u_{x_1}, u_{x_2}, u_{x_1 x_2}\}$. We also use the empty index $\phi$ for $u$, i.e., $u_\phi := u$. For simplicity of exposition we assume a set with a single PDE denoted $\mathcal{P}_u$ for a function $u(t, x)$ over a spatial dimensions $x := (x_1, \ldots, x_{n_d})$ with $t \in [0, t_f]$ and $x_i \in [0, x_{i,f}]$.

With these simplifications $\mathcal{P}_u$ has the following general form:

$$\mathcal{P}_u : \sum_{m \in \mathcal{M}} c_m(t, x; u_{\text{data}}) u_m = b(t, x; u_{\text{data}}), \quad (1)$$

where $\mathcal{M}$ is a multi-index set, $u_m$ are partial derivatives and $c_m, b$ are time-space varying coefficients over $\mathcal{D}$. Given a

set $\mathcal{B}$ of boundary coordinates we complete the PDE specification by specifying the initial and boundary conditions with a function $\omega$ defined over $\mathcal{B}$ as

$$u(i) = \omega(i), \quad \forall i \in \mathcal{B},$$

In general mechanistic PDE networks allow flexibility in the specification of initial and boundary conditions which may be of the Dirichlet (i.e., boundary conditions on $u$), Neumann (i.e., boundary conditions on partial derivatives) or mixed types and may be arbitrary linear constraints (subject to well-posedness). The time and space varying coefficients allow the PDE representations of mechanistic PDE networks to be very expressive and allow for modeling of complex dynamical phenomena. In this paper we assume that PDE representations are well-posed and that the PDEs together with the boundary conditions have a unique solution.

**Discretization.** To implement the PDE in a neural network we discretize the rectangular domain $\mathcal{D}$ over time and space. We assume one time and one spatial dimension. A given time interval $[0, t_f]$ is discretized into $n_t$ intervals $[t_i, t_{i+1}]$ with step size $s_{t_i} = t_{i+1} - t_i$. Similarly a spatial interval $[0, x_{i,f}]$ is discretized into $n_{x_i}$ intervals $[x_{i,k}, x_{i,k+1}]$ with steps $s_{x_{i,k}}$.

All functions $c_m, b, \omega$ produced by the mechanistic encoder are now assumed to be defined over the discretized spatio-temporal grid. With mechanistic networks the steps sizes can also learned and are then produced by the mechanistic encoder. All together the mechanistic encoder parameterizes the PDE by producing the coefficients, boundary parameters and the step sizes defined over the spatio-temporal grid for a given multi-index set $\mathcal{M}$.

$$[\{c_m\}_{m \in \mathcal{M}}, b, \omega, \{s_{t_i}\}, \{s_{x_i}\}] = f_{Enc}(u_{data}) \quad (2)$$

Next the generated PDE is fed to the NeuRLP-PDE solver which produces the solution and all partial derivatives specified in $\mathcal{M}$ as $\{u_m\}_{m \in \mathcal{M}}$. The output of the solver is fed to a task-specific decoder or directly used in a loss. All aspects of the PDE model in equation 1 including coefficients, initial and boundary conditions and step sizes are differentiable.

**Nonlinear PDEs.** The PDE representations generated by MechNN-PDE are linear, time-space varying. However, the model is not restricted to learning linear PDEs. Nonlinear PDEs can be represented by using nonlinear basis functions, either directly over the input data $u_{\text{data}}$ or by learning nonlinear functions of $u$. We illustrate this further in the section on discovery.

## 3. Linear PDE Solving With Differentiable Optimization

The workhorse of mechanistic PDE networks is NeuRLP-PDE – a specialized, parallel and differentiable solver for

linear PDEs. NeuRLP-PDE solves linear PDEs, with partial derivatives of arbitrary order, over a discretized spatio-temporal grid by reducing PDE solution to a relaxed differentiable constrained optimization (Young, 1961).

Briefly, the constrained problem specifies that 1) the discretized PDE holds over all points of the grid, 2) that the initial or boundary conditions hold and 3) that the computed solution is a smooth function over the grid, and the objective is to minimize approximation error. We then solve a relaxation of the optimization problem so that we can differentiate through the PDE solution to the PDE data using techniques from differentiable optimization.

To fully specify the constrained optimization, we specify the optimization variables and constraints. We consider the discretized Cartesian grid with $t$ and $x$ discretization $\{t_i\}_{i\in[n_t]}$, $\{x_j\}_{j\in[n_x]}$. Let $\mathcal{I}$ and $\mathcal{B}$ denote the set of grid and boundary indices, respectively. We also use the shorthand $\texttt{next}_c(i)$ and $\texttt{prev}_c(i)$ where $i \in \mathcal{I}$ and $c$ is a dimension for the next and previous adjacent grid points to point $i$ for coordinate $c$.

*Variables.* We are given a multi-index set $\mathcal{M}$ with $\phi \in \mathcal{M}$. We create variables $u_{\phi,i}$, $i \in \mathcal{I}$ for the solution $u$ at each grid point. We also create variables for each partial derivative for each $i \in \mathcal{I}$, obtaining variables denoted $u_{t,i}$ and $u_{xx,i}$ for partial derivatives $u_t$ and $u_{xx}$ and similarly for all other partial derivatives from $\mathcal{M}$. We denote the number of optimization variables by $n_v$.

The solver has three types of constraints: equation, smoothness and initial and boundary constraints.

*Equation* constraints specify that the left and right hand sides of the given PDE are equal over all grid points.

$$\sum_{m\in\mathcal{M}} c_{m,i} u_{m,i} = b_i, \quad \forall i \in \mathcal{I} \tag{3}$$

*Initial and boundary* constraints are specified for points in $\mathcal{B}$, where the form of the constraint can depend on the problem. Here we give Dirichlet-type conditions which specify the values of $u$ at points in $\mathcal{B}$.

$$u_{\phi,i} = \omega_i, \quad \forall i \in \mathcal{B} \tag{4}$$

We use central difference constraints to compute partial derivatives. We illustrate for a 3 point central difference approximation for a second order partial derivative $u_t$, with derivative multi-index $m$.

$$s_t^2 u_{m,i} = u_{\texttt{prev(i)}} - 2u_i + u_{\texttt{next(i)}} \quad \forall i \in \mathcal{I} \setminus \mathcal{B} \tag{5}$$

In practice we use 5 point central differences and use forward, backward finite differences at the edges of the grid.

*Forward and backward smoothness* constraints specify that the solution $u$ is smooth over the grid in each dimension in both directions. This is achieved by computing a 1D Taylor expansion at each grid point for each dimension and approximating the $u$ value at the next and previous adjacent

grids point for that dimension. The constraint specifies that the Taylor approximation and the solution variables at the next point should be close. We illustrate with a second-order Taylor approximation over the $t$ dimension.

$$u_{\phi,i} + s_t u_{m_1,i} + \frac{1}{2} s_t^2 u_{m_2,i} = u_{\texttt{next}_t(i)} \tag{6}$$

$$u_{\phi,i} - s_t u_{m_1,i} + \frac{1}{2} s_t^2 u_{m_2,i} = u_{\texttt{prev}_t(i)}, \tag{7}$$

where $m_1, m_2$ are multi-indices corresponding to the first and second partial derivatives of $t$.

We collect the variables $u_{m,i}$, $m \in \mathcal{M}$, $i \in I$ into a vector $z$. We can then write the constraints in matrix form as

$$Az = d, \tag{8}$$

with $A \in \mathrm{R}^{n_c \times n_v}$ being the coefficient matrix with $n_c$ constraints, and $d$ is the vector of constant coefficients.

We can solve eq (8) in a least squares sense by solving the corresponding normal equations.

$$A^\intercal A z = A^\intercal d. \tag{9}$$

Solving these equations gives a solution $z$ of the optimization problem and thus of the PDE. However, we also require the dual variables for the differentiating through the least squares optimization. For this we consider the following saddle-point formulation of least squares (Saad, 2003).

$$\begin{bmatrix} I & A \\ A^\intercal & 0 \end{bmatrix} \begin{bmatrix} \lambda^* \\ z^* \end{bmatrix} = \begin{bmatrix} d \\ 0 \end{bmatrix} \tag{10}$$

where $\lambda$ are the dual variables. This formulation can also be seen as the KKT constraints of the convex quadratic programming formulation of least-squares. Given the saddle-point formulation we can apply techniques from differentiable optimization for differentiating through the solution of constrained optimization problems (Amos & Kolter, 2017).

**Forward pass.** In the forward pass we solve the normal equations and compute the dual variables from 10 to obtain $z^*$ and $\lambda^*$.

**Backward pass.** Give scalar loss $l$ we consider the gradient, $g_z := \partial_{z^*} l$, relative to $z^*$. We require the gradient of $l$ relative to the matrix $A$ and vector $d$ which contain the optimization data. Following Amos & Kolter (2017), we can achieve this by solving the following saddle-point system:

$$\begin{bmatrix} I & A \\ A^\intercal & 0 \end{bmatrix} \begin{bmatrix} d_\lambda \\ d_z \end{bmatrix} = \begin{bmatrix} 0 \\ -g_z \end{bmatrix} \tag{11}$$

The gradient of the loss $l$ relative to $A$ and $d$ is $\partial_A l = d_\lambda z^{*\intercal} + \lambda^* d_z^\intercal$ and $\partial_d l = -d_\lambda$ (Amos & Kolter, 2017).

**Scaling PDE Solving.** The linear systems in equations (10), (11), (9) can become very large with increasing dimension of the PDEs resulting in very large grids over which the PDE is defined. As an example, for a PDE defined over 32x32 grids we have 5120 variables and 9212 constraints, requiring 377 MB of double precision dense storage. However for 32x32x32 (220k variables and 420k constraints)

we would need 780GB and for 64x64x64 we would need 50k GB. For such large grids the linear systems cannot be solved by dense methods. For the experiments we solve 1D PDEs with a dense solver since the memory requirement is reasonably bounded for small grids and batch sizes. However, for 2D or higher PDEs with midsize grids (say, 32 to 64 per dimension), the memory required for dense solvers is too great for GPU storage. The matrices, however, are highly sparse with 99.9% zero values for a 32x32 matrix and 99.999% zeros for a 64x64x64 matrix. For such problems we develop a multigrid preconditioner for use with a sparse iterative solver.

### 3.1. An Iterative Sparse Multigrid Solver

To make PDE solving feasible for more than two dimensions (including time) and for large grids, we build an iterative sparse solver for solving the normal equations. We use the FGMRES linear solver, the flexible variant of the Krylov-subspace linear solver GMRES (Saad & Schultz, 1986), as the base linear solver. As is well-known from numerical linear algebra, standalone iterative solvers can lead to poor solutions for complex problems and require *precondition-ing* for accurate solution. General purpose preconditioning methods such as incomplete LU or Cholesky factorization are not feasible for the large systems we encounter due to their high memory use and lack of GPU implementations. Instead, we build a multigrid *V-cycle* preconditioner for FGMRES. We briefly describe the multigrid V-cycle here and in Algorithm 1 in the appendix. More details on the multigrid methods and sparse iterative solvers can be found in Saad (2003) and Briggs et al. (2000).

**Multigrid V-cycle**. The multigrid *V-cycle* approach solves a linear system for a PDE on a succession of fine to coarse grids. Given a square linear system $Mx = b$ with solution $x^*$ and current solution $x_0$ we can compute the error as $\epsilon = x^* - x_0$ and residual as $r_0 = b - Mx_0$. We note that we can rewrite the linear system as $M\epsilon_0 = r_0$, where we wish to solve for the error $\epsilon_0$. Given an estimate for $\epsilon_0$ we can add it to $x_0$ to obtain an improved solution. Next we assume that we have a sequence of $n_g$ successive coarse-grained versions of $M$ (with the dimension of $M_{i+1}$, being, say, half the dimension of $M_i$) given as $M_1 = M, M_2, \ldots, M_{n_g}$.

In the first half of the V-cycle, at the $k$th grid the linear system $M_k\epsilon_k = r_k$ is partially solved by taking a few steps of a simple iterative *relaxation* method such as the Jacobi or Gauss-Seidel method, i.e., $\epsilon_k := \mathtt{relax}(M_k, r_k)$. Next the new residual $r'_k := r_k - M_k\epsilon_k$ in the linear system at grid $k$ is computed and transferred to the next coarser grid $k+1$ by a *restriction* operator designed for grid transfer, i.e., $r_{k+1} := \mathtt{restrict}(r'_k)$. The linear system at the coarsest grid $n_g$ is small and we solve for the exact error $\epsilon_{n_g}$ for the given residual $r_{n_g}$.

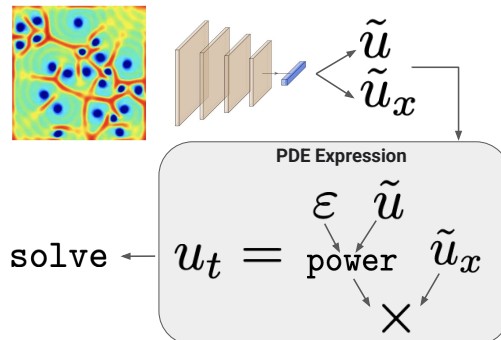

*Figure 1.* PDE expression building and discovery architecture. The expression is $u_t = \tilde{u}^\epsilon \tilde{u}_x$ with parameter $\epsilon$ as the exponent.

In the second half of the V-cycle the error at grid $k$ is transferred to the next finer grid $k - 1$, starting from the coarsest grid, by a *prolongation* operator, i.e., $\tilde{\epsilon}_{k-1} = \mathtt{prolong}(\epsilon_k)$. The prolonged error is added to the error at the finer grid, to obtain the new error at grid $k - 1$, i.e, $\epsilon_{k-1} := \epsilon_{k-1} + \tilde{\epsilon}_{k-1}$. The error is smoothed by relaxation $\epsilon_{k-1} := \mathtt{relax}(M_{k-1}, \epsilon_{k-1}, r_{k-1})$ The V-cycle ends when we return to the finest grid and we improve the solution $x_0$ by adding the estimated error, i.e., $x_0 := x_0 + \epsilon_1$.

In our solver we use the multigrid V-cycle as a preconditioner for FGMRES, which we found led to more accurate solutions compared with conjugate gradient methods and standalone V-cycle. We use Gauss-Seidel as the relaxation method and linear interpolation as the restriction and prolongation methods. To construct coarse grid matrices we use grid sizes with powers of 2 and successively halve each dimension whlist simultaneously doubling the step size in each dimension. We use size 8 for each dimension for the coarsest grid. The PDE coefficients and boundary conditions are coarsened by linear interpolation and the multigrid preconditioned FGMRES is applied to the matrix $A^\mathsf{T} A$ from the normal equations.

The resulting iterative solver is entirely sparse and batch-parallel on GPU for a batch of PDEs and is useful for higher dimensional PDEs where GPU memory cannot hold the full dense constraint matrices.

## 4. Mechanistic Networks for PDE Discovery

The *goal* of discovery is to find the equations governing spatio-temporal data representing some physical phenomenon. The *result* of governing equation discovery is a set of partial differential equations that, ideally, completely describe the given data. Further restrictions may be added to the form of the discovered equations so that the equations may be interpreted by human scientists. One such restriction is that the resulting equations should be composed only of *simple* functions. Another natural restriction is that the equations should *concise*, or sparse, with as few terms as

possible (Brunton et al., 2016).

In this section we present a Mechanistic PDE Network discovery architecture that allows discovery of simple and concise governing partial differential equations from spatio-temporal data. With Mechanistic PDE Networks the PDE representations can contain arbitrary differentiable expressions over inputs parameterized by neural networks as in Figure 1. This allows the design to model complex nonlinear PDE expressions, going beyond linear combinations of fixed basis functions.

### 4.1. Discovery Model

Given spatio-temporal data $u_{\text{data}}(t, x)$, where we assume two dimensions for simplicity, the discovery method generates a parameterized PDE *expression* along with initial and boundary conditions. $\mathcal{P}_u := \mathcal{P}_u(u, u_t, u_x, u_{tt}, u_{tx}, u_{tt}; \Theta)$. In general the form of the PDE may include arbitrary differentiable functions of any complexity, such as deep neural networks. However to satisfy the twin goals of simplicity and conciseness we restrict the allowed forms of the expressions appearing in PDEs to simple tree structures where each node is an elementary function with some weighting parameter. One common example of a tree expression structure in PDE modeling is a finite set of simple basis functions such as polynomials of a fixed maximum degree. Mechanistic networks, however, are not constrained to linear combinations of fixed basis functions and can represent more complex expressions by combining the data $u$ and its derivatives with parameters in a differentiable way (Figure 1).

Once the form of the expression is specified the aim of the discovery procedure is to find the appropriate parameters of the PDE expression given data. One could also imagine learning the structure of the expression itself; however, that is not a direction we take, and in this paper, we use fixed forms of PDE expressions.

**Parameterized PDE Expressions.** Given the restriction of conciseness, there must only be a small number of expression parameters that describe complex data. Furthermore, the initial values of these parameters might be far from the true parameters. If the PDE solution corresponding to the initial parameters is far from the true $u_{\text{data}}$ the iterative learning can become brittle and non-smooth due the small number of parameters available to the optimization process.

To make the learning smoother and more flexible, we parameterize the input to the PDE expression with a neural network. We do this by transforming the data $u_{\text{data}}$ to $\tilde{u} = \text{NN}(u_{\text{data}})$ with a neural network and compute the expression on the transformed data $\tilde{u}$ instead of $u_{\text{data}}$. This allows the learning to adapt both parameters and basis input to get to the target solution, leading to a more flexible learning procedure. A further advantage of parameterizing the

expression input is to make it robust to noise in the input data. To ensure that the final learned equation is correct for $u$, we also add a loss term $loss(u, \tilde{u})$ so that $\tilde{u}$ is close to $u$ at convergence.

**Discovery Example.** As an informative example, we illustrate the discovery method for Burger's equation: $u_t = uu_x + 0.1u_{xx}$, where the diffusion term, $u_{xx}$, has the fixed coefficient 0.1. Given $u_{\text{data}}$, we first compute a transformation $\tilde{u} = \text{NN}(u_{\text{data}})$. Choosing degree two polynomials as the expression functions we obtain the coefficients for $u_x$ and $u_{xx}$ respectively as $p_1(\tilde{u}) = \theta_0 + \theta_1\tilde{u} + \theta_2\tilde{u}^2$ and $p_2(\tilde{u}) = \phi_0 + \phi_1\tilde{u} + \phi_2\tilde{u}^2$, with $\theta_i$ and $\phi_i$ as learnable parameters. Choosing second order spatial and first order temporal derivatives we get a PDE expression of the form

$$u_t = p_1(\tilde{u}; \theta)u_x + p_2(\tilde{u}; \phi)u_{xx}. \qquad (12)$$

The true equation corresponds to the case where $\theta_1 = 1$, $\phi_0 = 0.1$ and the remaining parameters are 0. The initial and boundary conditions are obtained from $\tilde{u}$. In each iteration we solve Equation 12 using the NeuRLP-PDE solver obtaining solution $u$. We then compute the loss as a sum of the losses (either $L_1$ or $L_2$) between $u_{\text{data}}$, $u$ and $u$, $\tilde{u}$. We also include an $L_1$ loss over the parameters for sparsity. The total loss is computed as $l(u_{\text{data}}, u, \tilde{u}, \theta, \phi) = loss(u_{\text{data}}, u) + loss(u, \tilde{u}) + l_1(\theta, \phi)$.

The loss is iteratively minimized with gradient descent. Finally a concise PDE is generated by thresholding the parameters, with the ones below the threshold set to 0.

## 5. Related Work

The synthesis of machine learning and differential equations has been treated in a few different ways in the field. A major line of work approaches the problem from a machine learning perspective with supervised or unsupervised learning for partial differential equations (Li et al., 2020; Brandstetter et al., 2022). Another line of work in machine learning for scientific application aims at a synthesis of differential equation models with a *network-in-solver* approach by embedding neural networks in classical solvers (Chen et al., 2018; Rackauckas et al., 2020) enabling access to high quality numerical solvers in the context of machine learning. Physics-informed networks use a physics-aware loss to supervise network training (Raissi, 2018; Raissi et al., 2019). Mechanistic neural networks conversely are a *solver-in-network* approach that attempts to balance classical solvers and neural networks by combining restricted ODEs (Pervez et al., 2024; Chen et al., 2025) (and now PDEs), that allowing faster solving, with neural network learning together with techniques for handling nonlinear equations.

As an application of the synthesis of ML and differential equations various data driven techniques for discovering governing equations have been explored for ODEs such as

SINDy (Brunton et al., 2016), SINDy-PI (Kaheman et al., 2020) and PDEs including PDEFIND (Rudy et al., 2017) and weak forms (Reinbold et al., 2020; Messenger & Bortz, 2021) extending the SINDy framework. Physics informed networks (Raissi et al., 2019; Raissi, 2018) have also been used for inverse problems but unlike UDEs (Rackauckas et al., 2020) and SINDy do not handle unknown physics. PDE-LEARN (Stephany & Earls, 2024) combines features of physics informed networks and basis libraries for PDE discovery. MechNNs (Pervez et al., 2024) build a discovery model for ODEs which we enhance and develop PDE models that handle nonlinear equations and unknown dynamics.

## 6. Experiments

We demonstrate PDE discovery on a number of PDEs in two and three dimensions (including time), including complex dynamical data from the 2D Navier Stokes and reaction diffusion PDEs. We examine the robustness of the method to noise. We also consider PDE parameter discovery in the case where the PDE cannot be expressed as a linear combination of fixed basis functions. Table 1 lists the PDEs that we use to demonstrate our method, including their general form.

**Robustness to Noisy Data.** We test discovery in noisy settings by adding Gaussian noise of variance $\sigma^2$ to the data. The noise standard deviation $\sigma$ is chosen to be a ratio $\sigma_{NR}$ of the data root mean squared value i.e., $\sigma := \sigma_{NR}||U||_{RMS}$, where $U$ denotes the dataset and $\sigma_{NR} \in [0,1]$ is the noise ratio (Messenger & Bortz, 2021). In the experiments, we report the noise level as a percentage as $\sigma_{NR}.100\%$ noise.

**Evaluation Metrics.** We use two metrics for evaluating the quality of the discovered equation. The *true positivity ratio* is defined as TPR $= \frac{\text{TP}}{\text{TP+FN+FP}}$ (Lagergren et al., 2020) and measures how many terms have been correctly identified. We measure the maximum relative error of the true coefficients as $E_\infty(\xi) = \max_{j:\hat{\xi}_j^* != 0} \frac{|\xi_j - \hat{\xi}_j^*|}{|\xi_j|}$, with $\xi, \hat{\xi}$ being the discovered and true coefficients (Messenger & Bortz, 2021). $E_\infty(\xi)$ serves as the measure of accuracy of discovered coefficients.

**Datasets, experiments, baselines.** We perform experiments

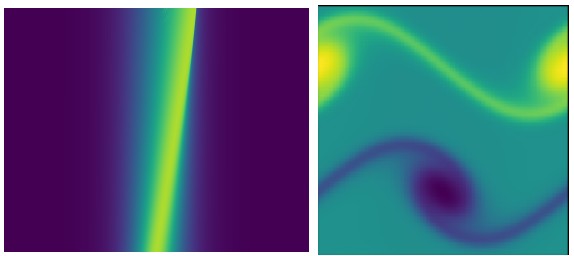

*Figure 2.* Inviscid Burger's Equation (discontinuous shocks are visible) (left) and the vorticity field data $\nabla \times U$ from an incompressible Navier-Stokes equation.

*Table 1.* PDEs for Discovery

| Name | Expression |
|---|---|
| Diffusion | $u_t = \nu u_{xx}$ |
| Viscous Burger's | $u_t + uu_x = \nu u_{xx}$ |
| Inviscid Burger's | $u_t + uu_x = 0$ |
| Porous Medium | $u_t = \nabla^2(u^m)$ |
| Ginzburg-Landau | $\partial_t A = A + \nabla^2 A - (1 + i\beta)|A|^2 A$ |
| Navier-Stokes | $\partial_t U + U \cdot \nabla + \nabla p = \nu \nabla^2 U$ |

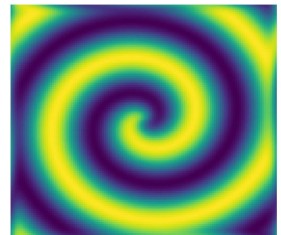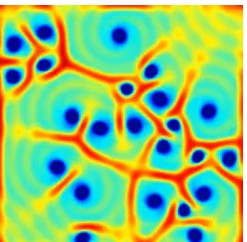

*Figure 3.* Easy case, showing $A^i$ (left) and harder, showing the norm $|A|^2$ (right) at a single time-step for reaction-diffusion data.

on a number of 1D and 2D (in space) PDEs chosen to cover a range of complexity. For the 1D equation, we demonstrate discovery with the diffusion equation and Burger's equation (both viscous and inviscid (i.e., without viscosity)). The inviscid Burger's equation is known to develop *shocks*, which appear as a discontinuity in wave propagation. We show that our proposed method can perform discovery in the presence of this form of discontinuity. The *porous medium* equation is a nonlinear equation that cannot be written in the form of linear combinations of fixed basis functions because of a real exponent. We show how the mechanistic network framework easily extends to such cases while many other methods cannot.

For 2D PDEs, we work with reaction-diffusion and incompressible Navier-Stokes equations. Ginzburg-Landau reaction-diffusion equations are frequently employed for modeling biological pattern development. We demonstrate discovery in two settings: 1) a relatively easy spiral dataset with a single spiral and 2) a harder setting with significantly more complex patterns where the baseline methods fail to recover the equations even in the absence of noise. We also demonstrate discovery for an incompressible Navier-Stokes example with a small viscosity term (Harpole et al., 2019). We set the viscosity term $\nu$ to 0.001 and show that our method is able to recover the equation while the baselines cannot. For baselines, we compare with PDEFIND (Rudy et al., 2017) and WeakSINDy (Reinbold et al., 2020), with implementations by de Silva et al. (2020), which are known to be robust to noise in many settings. We especially focus on comparing the robustness under noise of our method with the baselines.

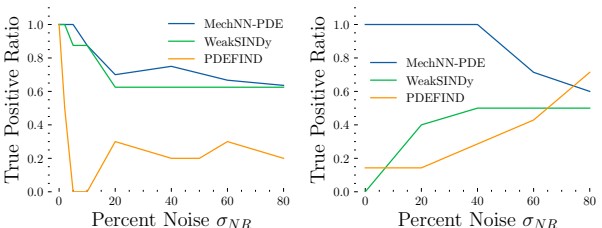

*Figure 4.* TPR for easy (left) and harder (right) reaction-diffusion experiments with varying amounts of noise.

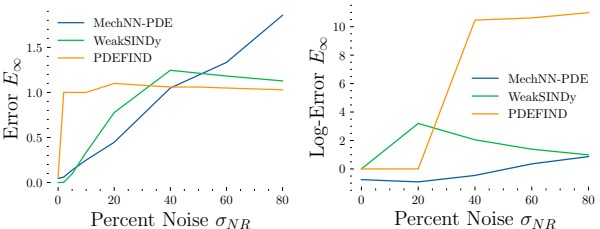

*Figure 5.* Log $E_\infty$ errors for the easy (left) and harder (right) reaction-diffusion experiment with varying amounts of noise.

### 6.1. Solver Unit Test

First, we verify that our solver is capable of solving some given linear PDEs. We test on the 2D Laplace PDE, $\nabla^2 u = 0$ on 32x32, 64x64, and 128x128 grids using the multigrid solver. We compare against a reference finite-difference implementation from the py-pde (PyPde) package with sinusoidal boundary conditions obtaining $L_2$ errors of orders $10^{-1}, 5 \times 10^{-2}$ and $10^{-2}$, respectively, showing the errors decreasing with increasing resolution. This simple evaluation shows that our solver is indeed capable of solving these equations to an accuracy comparable with traditional libraries while also being able to parallelize solving multiple equations on GPUs.

### 6.2. PDE Discovery

In this section, we demonstrate PDE discovery for equations in one and two spatial dimensions in addition to time.

**2D PDEs.** For two dimensional PDEs, we test on Ginzburg-Landau reaction diffusion equations and the incompressible Navier-Stokes equations. For 2D PDEs the PDEs are a set of coupled equations. For ease of training we demonstrate discovery on one equation from each of the coupled PDEs.

**2D Reaction-Diffusion Equations.** The Ginzburg-Landau reaction-diffusion equations are shown in Table 1 in complex form in the variable $A$. For ease, we write the complex equations as a system of two equations, where we take the first equation in our experiments.

$$A_t^r = D_1 \nabla^2 + A^r(1 - |A|^2) + |A|^2 \beta A^i \quad (13)$$

$$A_t^i = D_2 \nabla^2 + A^i(1 - |A|^2) - |A|^2 \beta A^r, \quad (14)$$

where $A^r$ and $A^i$ are the real and imaginary parts of $A$ and $|A|^2 = A^{r2} + A^{i2}$ and $D_1, D_2, \beta$ are parameters.

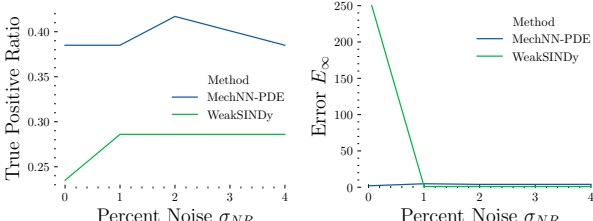

*Figure 6.* TPR and $E_\infty$ error for the Navier-Stokes dataset.

*Easy Case.* For the reaction-diffusion equations, we first test on an *easy* dataset with a single spiral pattern for which the baselines also perform well (Reinbold et al., 2020). For this example, the spatial domain has a length of 20 with equal spatial step sizes and a spatial resolution of 64x64 in the $x$ and $y$ dimensions. For time we use 128 time steps with a step size of 0.05. An example of the data at a single time step is shown in Figure 6 (left). The data was generated by a spectral method following the implementation in de Silva et al. (2020), setting $D1 = D2 = 0.1$ and $\beta = 1$.

*A Harder Case.* For reaction diffusion equations we also test on a more difficult dataset with more complex patterns, for which the baselines are unable to discover the correct equations even on noise-free data. The dataset has a spatial resolution of 256x256 over a Cartesian domain of size 10 with equal step size. The data was generated by using the Basilisk solver (Kenneally et al., 2020) starting from a random initial condition and running with a maximum time step size of 0.05. We set $D1 = D2 = 1$ and $\beta = 1.5$. We take only 128 time steps from the data for training. An example of the data at a single time step is shown in Figure 6 (right).

*Training Data and Parameterization.* We use the same training setup for both kinds of reaction-diffusion examples. For our model, the training data is mini-batched with a mini-batch size of 8. Each data example in the minibatch is of size 32x32x32, where the first dimension is time. The data is parameterized by 10-layer 2D ResNets, where we consider the time dimension as a batch dimension. The data consists of two scalar fields $A^r, A^i$, which are separately parameterized by neural networks before being used to build the PDE expression as $\tilde{u} = \text{NN}(A^r)$ and $\tilde{v} = \text{NN}(A^i)$.

*PDE Expression.* The PDE expression model uses third degree polynomial functions built from the parameterized inputs $\tilde{u}, \tilde{v}$ and second order derivatives in $x$ and $y$. We use four separate polynomial expressions in the PDE: one serving as the coefficients of the $u$ term, another two for the $u_{xx}$ and $u_{yy}$ terms, and one as the right hand side $b$. The polynomial basis parameters for discovery are computed by a two-layer MLP with a learnable input, which we find to improve training over a flat parameter vector. The boundary conditions are learned Dirichlet boundary conditions and are simply chosen to be the boundaries of $\tilde{u}$. Given the PDE expression and the boundary conditions, the PDE expression

is fed to the solver to get solution $u$.

*Loss.* The final loss is an $L_1$ loss, which we find improves discovery. The loss is composed of four terms. The first is a loss $l_1(u, A^r)$, minimizing the loss between solution and data. The other two terms are losses for the parameterizations $\tilde{u}$ and $\tilde{v}$ as $l_2(\tilde{u}, u)$ and $l_3(\tilde{v}, A^i)$. Since we only solve one equation at a time, we do not have a learned solution to the second coupled equation and $\tilde{v}$ minimizes the loss directly with the data $A^i$. The final term in the loss is a sparsity term, also $L_1$ for the basis polynomial parameters. We use a weight of $10^{-4}$ for the sparsity term. In particular, we note that we do not threshold the parameters or repeat the optimization in our experiments with thresholded parameters, using only a single training cycle. Repeating the optimization can and does improve the accuracy of the learned coefficients, but also increases training time, which is why we do not repeat the optimization for the results in this paper.

The model is trained with Adam with a learning rate of $10^{-5}$, which we fix for all experiments. We do not use learning rate scheduling for the experiments for simplicity and use a fixed learning rate throughout. We repeat the training with up to $80\%$ noise added to $A^r$ and $A^i$.

*Results.* The TPR results are shown in Figure 4 and the $E_\infty$ errors are shown in Figure 5 for the various quantities of noise for both the easy and hard experiments for MechNN-PDE, WeakSINDy and PDEFIND. For clean data in the easy example, we see that all methods recover the non-zeros terms with a TPR of 1 with small error in the coefficients.

Upon increasing noise levels, we find that PDEFIND worsens the most, and the TPR drops significantly to 0-0.2 for the easy case while the error becomes very large. WeakSINDy performs much better with noise on the easy dataset, and MechNN-PDE and WeakSINDy have similar TPR and error with noise except for $80\%$ noise, where WeakSINDy has better error for similar TPR as MechNN-PDE.

For the harder case, neither PDEFIND nor WeakSINDy is able to find any significant non-zeros terms even with clean data. The TPR seems to increase later on for PDEFIND, however, this is spurious due to the large error as seen in the plots. WeakSINDy also performs no better than random in this setting. MechNN-PDE, however, is able to maintain high TPR with noise for up to $40\%$ noise, with the TPR dropping to 0.6 at $80\%$ noise, showing consistent performance in TPR and error with noise.

**Incompressible Navier-Stokes.** We use the first component of the velocity $U_1$ from the coupled equations for discovery given as $U_{1t} + U \cdot \nabla U_1 + p_x = \nu \nabla^2 U_1$, where the pressure gradient $\nabla p$ is given. We use a similar setup for the reaction-diffusion equations, except that we use degree 1 polynomials. We include interaction terms with the first derivatives. We use a small viscosity $\nu = 0.001$ for the data

(Figure 2). A comparison of the results with WeakSINDy is shown in 6, where WeakSINDy is unable to discover the system. For MechNN-PDE, the TPR is still lower than 1 with low noise. On inspection of the coefficients, it appears that this is due to the presence of zero terms with small coefficients that are of the order of the viscosity which appears as false positives, even though the viscosity and non-zeros terms are accurately recovered.

**1D PDEs.** We demonstrate discovery on the 1D diffusion equation and inviscid and viscous Burger's equations. We use degree 4 polynomials for the coefficients of the $u_x$ and $u_{xx}$ terms. For all cases in the noise-free case the method discovers the true equation. For Burger's equation we experiment with noisy data and compare with WeakSINDy and find both methods to be on-par with each other. The results for the are shown in Figure 7 and Appendix A.

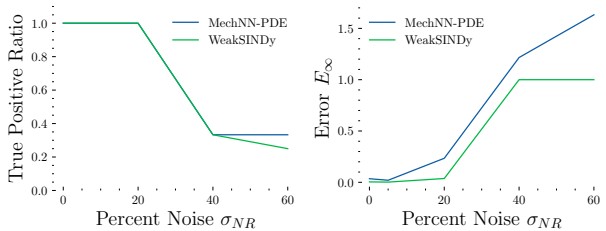

*Figure 7.* TPR and error for viscous Burger's equation. Comparing with WeakSINDy

### 6.2.1. DISCOVERY BEYOND GENERALIZED LINEAR

MechNN-PDE can represent complex differentiable expression in PDEs, going beyond expressions representable as linear combinations of fixed basis functions. We demonstrate with the task of parameter discovering for the Porous Medium equation for which the expression cannot be written as a linear combination of fixed basis functions. The porous medium equation is given as $u_t = \nabla^2(u^m), m > 0$, and we want to recover the parameter $m$ given data. We set $m = 2.675$ in the data and do not use any other basis functions. We note that $m$ is a positive real value which precludes fixed polynomial basis functions. The task is to discover the exponent of $u$. We build the PDE expression with an exponent for the parameterized data $\tilde{u}$. For this example we also parameterize the gradient with a neural network. With clean data we recover the coefficient $2.64\pm0.03$ close the true value of 2.675.

### 6.3. Multigrid Hyperparameters, Time, Memory

We show the effect of varying hyperparameters of the multigrid algorithm on solving the Laplace equation (Figures 8, 9, 10 in Appendix C). The hyperparameters varied are the number of FGMRES iterations for two resolutions, the number of V-cycles and the number of Gauss Seidel steps. We measure relative error and show decreasing error with

the number of iterations.

In Figure 11 (Appendix C) we show the time and memory used in solving the Laplace equation with varying resolutions.

## 7. Conclusion

This paper presents a new method for discovering governing PDEs from data. We build specialized multigrid parallel and differentiable solver for solving PDEs in batch which is then incorporated in a discovery architecture. Experiments show that the method can discover complex equations from data while being robust to noise and expands the class of PDEs that can be modeled by machine learning methods.

*Limitations.* Scalability is a main limitation of grid-based PDE methods and one way forward might be to explore a combination of grid-based and mesh-free techniques. Another shortcoming is that the method sometimes sacrifices accuracy for speed. This could be resolved by better multigrid solvers and faster GPU implementations. Finally, we do not consider the identifiability of the PDEs being discovered and the conditions under which discovery is possible. Extending the analysis of Yao et al. (2024) to PDEs would be an interesting starting point.

## Acknowledgements

*AP.* This project has received funding from the European Union's Horizon 2020 research and innovation programme under the 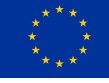 Marie Skłodowska-Curie Grant Agreement No. 101034413.

*FL.* This research was funded in whole or in part by the Austrian Science Fund (FWF) 10.55776/COE12. For open access purposes, the author has applied a CC BY public copyright license to any author accepted manuscript version arising from this submission.

## Impact Statement

This paper presents work whose goal is to advance the field of Machine Learning. There are many potential societal consequences of our work, none which we feel must be specifically highlighted here.

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

## A. 1D PDE Experiments

We test discovery on clean data on the diffusion equation with a diffusion coefficient of 0.01. We obtain a TPR of 1 and $E_\infty$ value of 0.224. We show results for the viscous and inviscid Burger's equations in Figure 7 and Table 2, comparing with WeakSINDy. We find both methods to perform similarly on this problem.

*Table 2.* Inviscid Burger's Equation for clean and 10% noisy data.

|  | MechNN-PDE | | WeakSINDy | |
| --- | --- | --- | --- | --- |
| Noise | TRP | $E_\infty$ | TRP | $E_\infty$ |
| 0 | 1 | 0.043 | 1 | 0.001 |
| 10 | 1 | 0.0020 | 1 | 0.012 |

## B. Multigrid V-cycle Algorithm

Algorithm 1 shows the multigrid V-cycle (see Saad (2003)) that we use to precondition the FGMRES algorithm. We use Gauss-Seidel as the relaxation method and linear interpolation as the restriction and prolongation operators.

---

**Algorithm 1** V-cycle

---

**Require:** `relax`, `restrict`, `prolong` operators
**Input:** $M_i, x_i, b_i; i \in [m]$
$x_i = \texttt{relax}(A_i, x_i, b_i)$
$r_i = b_i - A_i x_i$
$r_{i+1} = \texttt{restrict}(r_i)$
**if** $m == i + 1$ **then**
    Solve $A_m \delta_m = r_m$
**else**
    $\delta_i = \text{V-cycle}(A_{i+1}, 0, r_{i+1})$
**end if**
$\delta_i = \texttt{prolong}(\delta_{i+1})$
$x_i = x_i + \delta_i$
$x_i = \texttt{relax}(A_i, x_i, b_i)$

---

## C. Multigrid Hyperparameters and Memory

We show the effect of varying hyperparameters of the multigrid algorithm on solving the Laplace equation (Figures 8, 9, 10). The hyperparameters varied are the number of FGMRES iterations, the number of V-cycles and the number of Gauss Seidel steps.

In Figure 11 we show the time and memory used in solving the Laplace equation with varying resolutions.

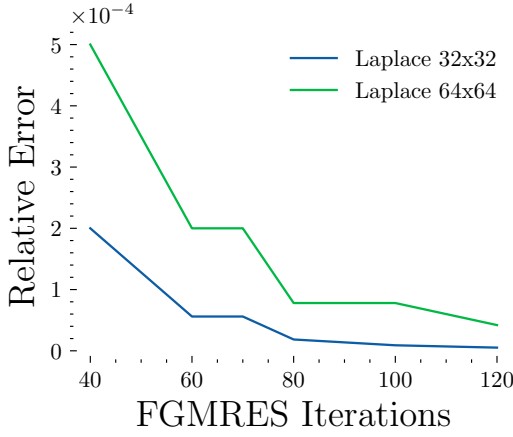

*Figure 8.* Relative error in linear system solving versus number of FGMRES iterations. The number of FGMRES restarts is fixed to 20. The number of preconditioning V-cycles is set to 1. Showing results for solving Laplace equation on 32x32 and 64x64 grids.

*Figure 9.* Relative error in linear system solving versus number of preconditioning V-cycles with FGMRES. The number of FGMRES restarts is fixed to 20 with 40 iterations. Showing results for solving Laplace equation on a 32x32 grid.

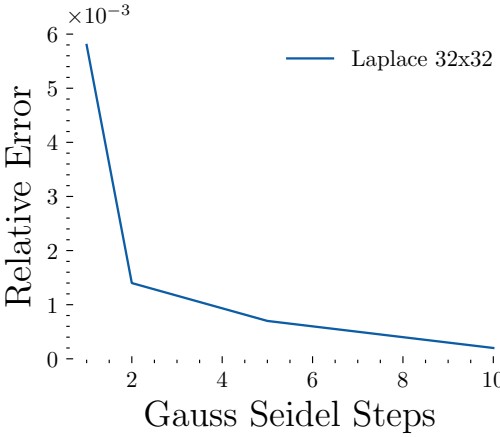

*Figure 11.* Time and GPU Memory usage for solving 2D Laplace equation for a batch size of 32 with multigrid preconditioned FGMRES.

| Grid | Memory (GB) | Time (seconds) |
| --- | --- | --- |
| 32x32 | 1.6 | 6.16 |
| 64x64 | 4.63 | 13.9 |
| 128x128 | 16.7 | 33.9 |

*Figure 10.* Relative error in linear system solving versus number of Gauss-Seidel cycles per preconditioning V-cycle with FGMRES. The number of FGMRES restarts is fixed to 20 with 40 iterations with 1 V-cycle per preconditioning V-cycle. Showing results for solving Laplace equation on a 32x32 grid.

