# OpenReview forum: "Mechanistic PDE Networks for Discovery of Governing Equations"
_ICML.cc/2025/Conference — ICML 2025 poster_

### Official Review · Reviewer_Zn4e · 2025-02-25

**Overall Recommendation:** 3

**Summary:**

This work extends mechanistic neural networks to discover PDEs. The approach relies on a numerical solver for PDEs, specialized for linear PDEs; however, the approach may still be applied to nonlinear PDEs by using nonlinear basis functions. Solutions are constructed as simple trees, which are restricted to be sparse and concise by Lasso regularization.

The method is as follows. Given data from some (perhaps set of) PDE(s) and a skeleton of the PDEs, an encoder produces a prediction of the solution $\tilde{u}$. Next, coefficients of a polynomial combination of $\tilde{u}$ are predicted and incorporated into the PDE structure. The constructed PDE is solved using the NeuRLP-PDE solver to compute $u$. Finally, a loss is calculated between the data and $u$, as well as $u$ and $tilde{u}$, and backpropagated through the solver and encoder.  The authors introduce a sparse solver with efficient implementation on GPUs, which serves as a backbone for the proposed approach.

**Claims And Evidence:**

The authors claim that the PDE solver used in this work is fast and efficient, as well as scalable. However, there are no ablation studies which show to effectiveness of this solver in the proposed approach as compared to other differentiable solvers.

Furthermore, the performance of the approach is not clear. The authors provide plots of TPR, but do not provide any details on the obtained solutions, for example, a table which shows the obtained parameters and solutions for the different test cases. This makes it difficult to judge the results.

The authors also claim that the combination of the machine learning approach with the numerical solver improve robustness, while the plots in Figures 4 and 5 seem to show similar performance with strictly machine learning based approaches. This claim seems difficult to accept given the limited evidence.

**Essential References Not Discussed:**

N/A

**Experimental Designs Or Analyses:**

Adam optimizer is used with a small learning rate and no scheduler is used. Is this the case across all baseline models as well? If so, this is a bit questionable, as learning rate scheduling and tuning can be critical to many applications.

**Methods And Evaluation Criteria:**

The experiments make sense, but the baseline methods seem limited. A recent approach, PDE-Learn, is referenced in this work, but not selected as a baseline. Is there a reason for this?

The evaluation of the approach for the experiments is also relatively limited. TPR and a maximum relative error over the coefficients are provided, but the relative performance of the solution to the discovered PDEs as compared with a numerical solution to the true PDEs is not given. Overall, the evaluation metrics make it difficult to determine exactly how well each approach is doing. Additional figures and organization of the results would be helpful to illustrate the ability to discover the PDEs. Since efficiency is also a central claim, the computation time and required computational resources of all approaches (including baselines) are also a critical point for discussion.

**Other Comments Or Suggestions:**

I would suggest the follow major changes for the work to improve the clarity:
- The paper focuses on PDE discovery, when the sparse solver is ultimately the main contribution in this work. I believe by first restructuring the paper to discuss the numerical solver and its contributions and advantages over existing works, the impact of this work would be more clear.
- Following this, MechNNs can be introduced. Currently, both section 2 and 4 discuss MechNNs, and so this information is split. It should be unified so that the reader can more easily follow the goals of the experiments section.
- This will naturally lead into the results section. Results for both the NeuRLP solver and MechNN-PDE can be shown here, and some details on the baseline methods can also be included.
- Additionally, the results section should be expanded to include more details. I think the previous sections could be shortened to cover the fundamental aspects of the approach, but technical details can be moved to the appendix.

I would like to ask the authors to review the work for typos as well.

Also, the NeuRLP acronym is not defined.

Line 342, right column, references Figure 6, when it should reference Figure 3.

**Other Strengths And Weaknesses:**

Strengths
1. The systems considered in this work are relatively challenging
2. The authors discuss limitations of their approach
3. The proposed numerical solver seems to be robust, with the ability to solve many systems of PDEs

Weaknesses
1. As previously mentioned, the baseline methods seem to be relatively weak
2. It is relatively difficult to draw comparisons between the performance of all approaches explored in this work and the evaluations are limited
3. The contribution of this work to the literature seems minimal in the paper's current form, and the claims made by the authors are not strongly supported by experimental evidence
4. In general, I find the structure of the paper to be hard to follow

**Questions For Authors:**

1. Are all the models retrained for each different noise setting? If so, how do the results compare with a model trained on a noiseless setting and evaluated on the noisy cases?

2. How is the threshold chosen? What if a PDE has many small parameters?

3. Can the approach be extended to the non-Cartesian domain?

4. The structure of the PDE is defined a priori, as this problem is usually very ill-posed. Therefore, it makes sense to only consider solving for the parameters of the PDE. Recently, several papers on finding analytical solutions to PDEs from data have been proposed. Would it be possible to extend this approach to find analytical solutions and their parameters?

**Relation To Broader Scientific Literature:**

Differentiable solvers are useful for a variety of tasks, particularly in solving inverse problems and engineering design. Fast, scalable solvers of this type are crucial, and this paper aims to address that, as well as show an interesting application in inverse problems (PDE discovery from data).

Additionally, many areas of science rely heavily on modeling by PDEs, for example weather dynamics. In weather prediction, these PDEs are often not known exactly, and come from heuristics and experimental testing. Likewise, weather data is often noisy. An approach such as this one will aid understanding in these fields, and allow scientists to construct more accurate models of physical phenomena.

**Theoretical Claims:**

No theoretical claims are made here.

---

> ### Author Rebuttal · Authors · 2025-03-31
>
> We appreciate the reviewer’s comments.
>
> **Main Contribution and Paper Structure**
>
> We would like to strongly emphasize that we do not claim to replace or improve existing PDE solvers. Our contribution is to enhance the MechNN model (Pervez et al, ICML 24) that works with ODEs to support PDE representations, together with an efficient way of solving the representations and applying the model for PDE discovery. The PDE solver we develop is specialized for this architecture, where the PDE terms are produced by a (non-smooth) neural network output and solves a relaxation. The sparse solver is intended *to make MechNN feasible for PDEs* with multiple dimensions.
>
> Furthermore, we make no claims of either speed or memory improvement over the baseline sparse regression methods which are not neural networks and require much fewer resources. We do claim that our model is more expressive (handles complex expressions) and can handle more complex data as shown in the discovery experiments.
>
> We are open to restructuring the paper for clarity as long as it correctly emphasizes our main contributions and will certainly consider the reviewer’s suggestions.
>
> **Plots look similar**
>
> We disagree with the reviewer’s assessment of the plots. We claim significant improvement on the more complex reaction diffusion system (Figures 4,5 right) over the baselines which entirely fail for this dataset.
>
> We note that the TPR and max coefficient error plots should be considered together, since the correct terms have to be discovered for the coefficient error to be meaningful. The baselines fail in all cases to discover the correct terms (with low error) including the case with no noise. For our model both TPR and coefficient error show gradual degradation with noise, with near perfect recovery in low noise settings even though we do not fine-tune after thresholding.
>
> On the easier reaction diffusion system the WeakSINDy baseline performs well (although PDEFIND fails with noise) and our method matches its performance, with only some decrease in coefficient accuracy at the 80% noise level.
>
> **Evaluation/Simulation**
>
> The metrics of TPR and max coefficient error are chosen from the literature. In our view they provide a more quantitative and objective basis for comparison of discovery methods. Raw equations allow limited insight, especially when exact discovery is not possible in noisy settings. However, we will consider adding examples of discovered equations in the updated draft.
>
> Similarly, simulating a discovered equation does not serve as an accurate determinant of how well an equation’s form has been learned, especially in noisy settings. Nevertheless we link a video example of simulation for the harder reaction diffusion example simulating the ground truth and discovered equation on clean data (uploaded here: https://filebin.net/wqv4t3oki732nuq0). We will consider adding more examples in the next draft.
>
>
> **Burger’s equation**
>
> With the inviscid Burger’s equation there is only one term to be discovered with the coefficient of 1. However, since we do not finetune after thresholding there can be occasional variation in the discovered coefficients. The learned coefficient in the noise-free case for this experiment is 0.957 which gives a relative error (|1-0.957|/|0.957|) of 0.04. For the noisy case the coefficient is 1.01 with a relative error of 0.001. Running the experiment with fine-tuning after thresholding gives a small relative error (~0.001) for the first case as well.
>
> **Recent Methods**
>
> Due to lack of space, please see our response to reviewer Ct17 for details of the comparison with PDE-LEARN.
>
>
> **Hyperparameters**
>
> We clarify that the baselines (PDEFIND, WeakSINDy)  are sparse regression methods which do not employ any neural networks and therefore the learning rates etc, do not apply to the baselines. For our method, for simplicity, we fixed the learning rate across all our experiments and do not use any scheduling.
>
> **Retraining**
>
> Yes, the models have to be retrained for new data.
> In the discovery setting we consider, the PDE parameters to be discovered are global parameters and are not functions of the data. This is also true for the baselines. This setting does not allow direct evaluation of parameters on a new dataset. However, developing a setting where this is sensible is an interesting question for further work.
>
> **Threshold**
>
> We chose the threshold per PDE qualitatively, for simplicity. The choice of threshold can affect small parameters which is also the reason why the TPR in the Navier-Stokes example is less than 1.
>
> **Analytical solutions**
>
> Most PDEs don’t have analytical solutions, but perhaps the question is with regard to learning the expression structure?
> If so, then yes, we think that symbolic regression methods could be combined with this method to learn expressions together with parameters and we are considering approaches for this.
>
> We would like to thank the reviewer again for the review.

---

> > ### Comment · Reviewer_Zn4e · 2025-04-02
> >
> > I would like to thank the authors for taking the time to respond to my questions and concerns. Based on this response, it seems the role of the PDE solver may be overstated within the main text. From my perspective, this is framed as a main contribution, e.g. "The workhorse of mechanistic PDE networks is NeuRLP-PDE – a specialized, parallel and differentiable solver for
> > linear PDEs."
> > As I understand it now, the main contribution of this work is the extension of MechNNs from ODEs to linear PDEs, and not the solver. Nonetheless, the answer to the question "how is this extension performed?" seems to be that the ODE solver from the founding MechNN work is replaced with this PDE solver. Although perhaps a bit circular, I have no problem with this argument in principle; however, in this case, impressive results are required to make up for the limited new work in the approach. Although the authors claim that significant improvement is present, I only see this clearly in one case (hard reaction-diffusion) and some moderate improvement in the NS problem. As it stands now, I believe that the presentation of the work, both contributions and results, is still quite difficult to understand. For that reason, I will not change my overall evaluation, but I encourage the authors to further refine this work, as I believe there is potential in it.

---

> > > ### Author Response · Authors · 2025-04-02
> > >
> > > We thank the reviewer for the response. We would like to reiterate the following.
> > >
> > > **Contribution**
> > >
> > > The PDE solver remains a main contribution. Since that is the component that makes the extension of the MechNN ODE architecture to PDEs possible. It is also not a trivial addition since significant computational challenges have to be met to make the extension from ODEs to PDEs with multiple dimensions feasible. We solve this challenge by developing a specialized sparse multigrid PDE solver that works with differentiable optimization.
> > >
> > > **Evaluation**
> > >
> > > We believe that we have given ample evidence for superior discovery performance for our model. In summary
> > >
> > > * We demonstrate significant discovery on a complex reaction-diffusion and Navier-Stokes equation, where the baselines completely fail in both cases. We show that we can perform robust discovery in the presence of noise in these equations. We present this with quantitative evaluation metrics.
> > >
> > > * We also show that our method can model PDEs with complex expressions such as the Porous Medium equation. This equation cannot be modeled by the baselines since they are limited to linear combinations of fixed basis functions.
> > >
> > > * In addition to this we show that we match the performance of the baselines on simpler problems including simpler reaction-diffusion, diffusion and Burger’s equations (both viscous and inviscid).

---

### Official Review · Reviewer_coX1 · 2025-03-10

**Overall Recommendation:** 4

**Summary:**

This paper presents a model that learns spatial-temporal PDEs from data samples. The model selects a set of basis functions with spatially and temporally varying coefficients over the problem domain. Next, it implements a multigrid solver to solve the proposed PDE. Finally, the model learns a PDE by backpropagating the gradients and optimizing a loss function on data terms. The paper demonstrates the method’s efficacy on classic 1D and 2D PDE instances.

## update after rebuttal

Thank you for your rebuttal. I didn't have too many questions to ask in my initial review, and the rebuttal didn't greatly affect my overall evaluation of this work. I will maintain my current score.

**Claims And Evidence:**

Most of them look OK to me.

I have one comment on the claim of “discovering” PDEs from data. I notice that the experiments generated data from known PDEs. I understand that having a ground-truth PDE is good for calibrating the efficacy of the “discovery” of PDEs, and many prior works used this setting as well. However, I feel it would be more proper to use the term “discovering” PDEs if the data are from a (real-life) experiment without a known PDE. Perhaps “recovering” or “reconstructing” a PDE is a more appropriate term for what is going on in this paper. I understand that this might be an unpopular opinion, and I won’t hold this against the paper.

**Essential References Not Discussed:**

Looks good to me. I don’t have extra references to suggest.

**Experimental Designs Or Analyses:**

I read the experiments, and I think the results look reasonable.

I have less experience with the baselines in the paper, so I am not sure whether they are the SOTAs for comparisons and will let the other reviewers decide. I also wonder whether PINNs and their variants are baselines for this work and would like to hear the authors’ thoughts on them.

**Methods And Evaluation Criteria:**

Most of them look good to me. In particular, I want to commend the effort to build a multigrid solver. The authors definitely deserve some credit for this.

I don’t quite get the neural network mapping (line 267) and would like to suggest an ablation study on it.

**Other Comments Or Suggestions:**

A minor comment: Line 95: is the notation x1x1 a typo? Based on the context, x1x2 seems to make more sense.

**Other Strengths And Weaknesses:**

I want to second the multigrid aspect of this work. Quite a few previous works evaluate their methods on small-sized toy examples only. I highly appreciate that this paper is willing to implement a multigrid solver in this setting, which has the potential to scale up the problem size.

On the negative side, the problem size (256 x 256?) in this work is still relatively small compared with what modern multigrid solvers can achieve (i.e., solving linear systems with > millions of variables). I feel the current method hasn’t unleashed the full power of multigrid solvers, and I am curious to know which part in the method is now the bottleneck that prevents it from being applied to large-scale problems.

**Questions For Authors:**

None for now. I look forward to a constructive reviewer-author discussion.

**Relation To Broader Scientific Literature:**

I think the proposed method overall is new and interesting. I can think of some related works that tackle similar problems or use similar techniques, none of which can challenge the novelty of this paper.

**Theoretical Claims:**

N/A.

---

> ### Author Rebuttal · Authors · 2025-03-31
>
> We thank the reviewer for the comments.
>
> The reviewer’s point regarding use of the term ‘discovery’ is well received. However, the term is now endemic in the literature and we chose to employ the same term.
>
> We would like to appreciate the reviewer’s recognition of the effort that went into building the solver. We will certainly release the source code for our models with the camera ready version of the paper.
>
> **Multigrid**
>
> It is quite correct that multigrid solvers are also used for higher resolutions. Our use case is different in a few respects from the usual use of multigrid.
> *  One difference is that we solve for the saddle point systems in equations 10,11 and we have extra constraints for ensuring smoothness which are usually not present in solvers.
> * Secondly, we work in a learning context with a nested iterative loop. The inner loop corresponds to the multgrid iteration and the outer loop is the learning loop that can run for hundreds of epochs.
> * Similarly we work in a batch setting where in each inner iteration we solve a minibatch of PDEs, all with their separate multgrid steps.
> * Furthermore the backward pass also uses the batched multigrid solver which doubles the memory use.
> * It is known that multigrid performance deteriorates with discontinuous coefficients. The data produced by neural networks over the grid (especially early in training) can be non-smooth which can deteriorate multigrid performance.
> * Our current implementation is in Pytorch which is not ideal for iterative processing. A native CUDA implementation can potentially produce significant computational improvements, but requires further work.
>
> Very high multigrid resolutions would have an adverse impact on the walltime, making the method very slow. Nevertheless, different from standard multigrid approaches, since we also have a learning component which can infer the underlying PDE given only a (batched) portion of the full resolution data. This implies that the solver is not required to solve for the full resolution data in each step. Rather it only has to solve for the given smaller resolution patch of data making it more efficient.
>
> **Encoder NN**
>
> The neural network transform (line 267) allows parameter capacity to the model to make it flexible. Without the transformation the only learnable parameters would be the PDE parameters, parameterizing the expression, which are few in number. This can lead to a brittle model making it harder to recover from bad initialization or noise. We will consider adding an ablation in the next draft.
>
>  $x_1 x_1$ should be $x_1 x_2$. Yes, that is a typo in this case.
>
> We would like to thank the reviewer again for the review.

---

### Official Review · Reviewer_Ct17 · 2025-03-14

**Overall Recommendation:** 4

**Summary:**

The paper presents mechanistic PDE networks, a method for discovering PDEs from spatiotemporal data. The proposed method integrates a differentiable PDE solver into the neural network and uses the neural network predictions to model the PDE instead of the raw data. The method discovers PDEs using the spatiotemporal data by expressing the PDE as a learnable polynomial kind of ansatz. The paper applies the proposed method to discover popular PDEs in the neural PDE solver domain and compares them with WeakSINDy and PDEFIND methods.

## Update after rebuttal

The paper presents advancements in discovering equations from data. The paper also presents the limitations of baselines and mitigates the challenges therein for 2D problems. Hence, I have raised my score considering the author's responses.

**Claims And Evidence:**

The claims in the paper regarding applicability of the proposed method for noisy data is validated through experiments.

**Essential References Not Discussed:**

References are adequate.

**Experimental Designs Or Analyses:**

Yes, I checked for all provided numerical experiments

**Methods And Evaluation Criteria:**

The experimental setting regarding dataset and noisy data draw similarity with conventional methods employed for PDE discovery.

**Other Comments Or Suggestions:**

Typo:

In line 95, should the product be taken for two different spatial coordinates?

Suggestions:

The authors might consider including a discussion on how the proposed method is advantageous over using rational neural networks for PDE discovery.

**Other Strengths And Weaknesses:**

Strengths:

The paper presents a novel method for discovering PDEs using spatiotemporal data, where the derivative computations do not need to be performed on the data but are instead performed on the neural network outputs.

The experiments show that the proposed method is robust to noise and outperforms WeakSINDy and PDEFIND.

The proposed method is scalable and the polynomial kind of ansatz enables generalized PDE operators.

Weaknesses:

The comparison with baseline methods is restricted to specific methods, and the paper does not compare it with recent methods.

The memory requirements increase with an increase in grid size. This is problematic for high-dimensional PDE discovery.

The computational cost of the method has a trade-off with accuracy, as also mentioned by the authors in the paper.

**Questions For Authors:**

The paper does not consider PDEs under external acting forces and only showcases their applicability for autonomous systems. How would the proposed method be applied to discover the PDE in case of unknown and known external acting forces?

How does the method guarantee the uniqueness of the identified PDE? How will the method behave when multiple PDEs govern the data equally well?

Extending the proposed method to discover stochastic PDEs or fractional differential equations seems complicated. How does one consider in priori the correct dynamical system framework modeling the dynamics?

It would be informative if the authors compare the proposed method with the rational neural networks approach of discovering governing PDEs, for instance, as presented in PDE-LEARN.

What will be the limitations and failure modes of the proposed method?

**Relation To Broader Scientific Literature:**

Discovering of governing equations from data is an important topic in many field of science and engineering. The proposed AI method is in the direction to model these problems motivated through canonical problems.

**Theoretical Claims:**

Not applicable

---

> ### Author Rebuttal · Authors · 2025-03-31
>
> We thank the reviewer for the comments. We attempt to address the concerns raised below.
>
> **Non-autonomous systems**
>
> Yes, in this paper we focus on autonomous systems. However, non-autonomous systems can be similarly handled since the method allows spatiotemporally varying terms. Any known external forces can be represented as time-space dependent terms similarly to what is achieved with basis functions. Any unknown forces that cannot be simply represented can be represented by a neural network. This is possible because the method allows arbitrary differentiable expressions in PDEs.
>
> **Identifiability**
>
> Whether the method finds the unique PDE requires further investigation into identifiability of the PDE learning problem. Identifiability is something we do not deal with in this paper and assume that the problems are identifiable and that there is a unique governing equation.
>
> We have not considered stochastic or fractional differential equations thus far and it would be interesting to consider whether such types of equations can be modeled in this way.
>
>
> **PINN style methods**
>
> The PDE-LEARN approach extends the PINN approach to PDE discovery. It reduces a PDE residual, constructed using basis functions, at random collocation points together with approximate L0 norm regularization. One difference of this approach with MechNN is that MechNN has a stronger physics informed bias due to the explicit modeling of PDEs as constraints.
>
> Unfortunately, the PDE-LEARN release does not have support for coupled equations, so it cannot be used with our 2D experiments in its current form. The main equations that we focused on were the 2D reaction diffusion and Navier-Stokes PDEs which are coupled equations.
>
> Nevertheless, we trained PDE-LEARN on our 1D Burger’s equation dataset with a 1/cosh(x) initial condition. Despite repeated attempts with multiple settings of hyperparameters, we found it to always over or under prune the equation (even on clean data), performing worse than the SINDy baselines from our paper. The maximum TPR we obtain for this experiment is 0.5, compared with a TPR of 1 for the SINDy baselines and MechNN.
>
> In particular one advantage of our method is that the modeled PDEs can contain arbitrary differentiable expressions and are not limited to linear combinations of fixed basis functions. Many other methods including PDE-LEARN model PDEs as linear combinations of fixed basis functions. A concrete example is the porous medium equation for which the equation cannot be modeled in this way, whereas we show that we can recover the true parameters.
>
> **Limitations**
>
> The following are some limitations of the method.
>
> * The method is slower than the baseline sparse regression methods.
> *  The method is currently limited to Cartesian grids.
> * Although the method can represent arbitrary differentiable expressions, there remains the need for theory that dictates the conditions under which complex expressions can be exactly identified.
> * There is a tradeoff between accuracy and speed in the multigrid solver which needs to be tuned.
> * Application to very high dimensional data is not feasible in the current form.
>
> We thank the reviewer again for the review and hope that we were able to meet any concerns.

---

> > ### Comment · Reviewer_Ct17 · 2025-04-06
> >
> > Thanks for the detailed response. The authors have addressed my concerns, and I have raised the score to 4. In the camera-ready version, it would be helpful to include the limitations and the additional experiment discussed here to foster further research.

---

### Official Review · Reviewer_2aLy · 2025-03-21

**Overall Recommendation:** 3

**Summary:**

The paper introduces a new methodology for learning PDEs on data. The key contribution is an optimization of how partial derivatives are handled in the network. Firstly, the theoretical formulation includes a dual formulation that elides a way to backpropagate through a linear solve effectively. Secondly, they introduce an optimized linear solving using a multigrid algorithm implemented on GPU. The paper then demonstrates that their method learns a few test PDEs more accurately than baseline PDE discovery methods with various levels of noise.

**Claims And Evidence:**

There are three distinct dimensions that the paper improves:
- Flexibility by allowing nonlinear backpropagation with PDE terms embedded.
- A very fast and efficient GPU implementation of a multigrid V-cycle algorithm for linear solving.
- Improved accuracy  on the PDE learning task.
The paper only demonstrates evidence of the improved accuracy.

A key improvement purported by this paper is supposedly the scaling and optimization of the new solver. However, no performance metrics or comparisons are reported. There is only theoretical calculations of memory utilization. It would greatly strengthen the paper to show speed and memory usage comparison with the baseline methods, as a function of the domain size.

**Essential References Not Discussed:**

Are there more papers about multigrid linear solvers implemented on GPU, even looking beyond the ML literature into scientifc computing? Is this the first time a GPU implementation of the multigrid method was published?

**Experimental Designs Or Analyses:**

I did not see any issues in the comparison between PDE learners sound.

As mentioned, there could be more elaboration on the solver efficiency. There is a brief discussion of solver convergence in Section 6.1, but this does not have any data nor figure references. It is great to discuss testing methodology at the forefront, but this leaves some questions: What is meant by errors decreasing with increasing grid size? Shouldn’t it go the other way around? What is the convergence rate? Is this the extent of the testing of the code? What is the convergence rate in terms of iterations of the V-cycle solver?

**Methods And Evaluation Criteria:**

As mentioned, evaluation of the purported claims of a fast solver are missing.

The results are also missing evidence of the discovered PDE forms or failure to discover the forms; see line 418 on page 8.

**Other Comments Or Suggestions:**

- Page 2: “Derivatives never have to be directly on data and are” Firstly, is there a mistake in this sentence? Secondly, what does it mean? Which derivatives? Do you mean parameter tangents, or partial derivatives?
- Page 2: Is the discretization only done for 1D? The paper has 2D problems: could you write the math in a higher dimension instead?
- Page 8: “We note that m is a positive real value which precludes..” What is that supposed to mean?
- Define FGMRES and GMRES for the audience.
- I don’t understand the paragraph headings that are bold versus italic on Page 5 and 7,8. It is not obvious which is higher or lower.

**Other Strengths And Weaknesses:**

I am supportive of the paper, but it is missing more discussion about the advantages of the solver technique. The paper is not sufficiently demonstrating the scaling and speed improvements. It mentions the theoretical scaling, but does not show the application to problems that the other PDE learning techniques cannot handle. As mentioned above, this would strengthen the contribution of the paper. That aspect would have the potential to be applied to other PDE discovery methods as a drop-in module, and even be adapted to other ML problems requiring a similar operation.

I also think that the method also has fewer constraints than the baseline methods: whereas e.g. WeakSINDy does a sparse linear regression, MechNN-PDE is doing backpropagation over the nonlinear expressions.

**Questions For Authors:**

- What aspects of the multigrid V-cycle section are new in your paper, on page 4? Did you propose any changes to the algorithm, or is this just the standard V-cycle?
- Do you have to solve the backward pass in Equation 11 with the multigrid solver as well?
- Is the FGMRES solver also your implementation, or do you integrate with another library? Which library, or how did you implement it?
- A few more details on the GPU implementation would be good for the paper, too. E.g., did you hand write CUDA kernels? For which library?
- Page 5: “Finally a concise PDE is generated by thresholding the parameters...” How is that done? Is the thresholding applied iteratively in a loop with parameter optimization, as in SINDy?  What is the thresholding parameter used?
- Page 7: “The data is parameterized by 10-layer 2D ResNets”: What does this mean? Which data, which parameterization? How does that fit in? What are the details of the architecture? Are they CNNs?
- Page 8, paragraph on line 411: Are the methods able to discover the equation forms here?
- The usage of the neural network for $\tilde{u}$ in the MechNN-PDE architecture is unclear. Would it be possible to include it in Figure 1?

**Relation To Broader Scientific Literature:**

As mentioned, the novel way of representing PDE solving in nonlinear networks coupled with a very efficient solver will be a contribution to the literature. Such an algorithm could be dropped in to many other PDE learning methods beyond the proposed MechNN-PDE.

**Theoretical Claims:**

I did not see any issues in the theoretical formulations in the paper.

---

> ### Author Rebuttal · Authors · 2025-03-31
>
> We appreciate the detailed review.
>
> **Clarification of our contribution.**
>
> We wish to clarify the nature of our contribution. We do not claim to replace or improve existing PDE solvers. Our contribution is to enhance the MechNN model (Pervez et al, ICML 24) that works with ODEs to support PDE representations, together with an efficient way of solving the representations and applying the model for PDE discovery. The PDE solver we develop is specialized for this architecture, where the PDE terms are produced by a (non-smooth) neural network and solves a relaxation. The sparse solver is intended to make the MechNNs *feasible* for PDEs with multiple dimensions.
> That said, we do not discount the possibility that the method could be useful for other applications.
>
> Furthermore, we make no claims of either speed or memory improvement over the baseline sparse regression methods which require much fewer resources. We do claim that our model is more expressive (handles complex expressions) and can handle more complex data as shown in the discovery experiments.
>
> **Missing Evidence and Failure.**
>
> We provide quantitative evidence of discovery performance using the TPR and Error metrics from the literature. A TPR of 1 means exact discovery of terms. This is shown in Figures 4,5 and 6 for our method and the baselines. The plots also show partial failure to discovery when noise is added. For instance with 80% noise (Figure 4) we only see a TPR of 0.6 which indicates extra or missing terms are present.  From Figure 6 we see that the Navier-Stokes equation is also not exactly discovered (TPR ~0.8). We present quantitative metrics since they allow more objective comparison than qualitative comparison between equations. We will include examples of discovered equations.
>
> **Solver Eval**
>
> We have performed further evaluation of the multigrid linear solver including time and memory usage, convergence of relative error with the number of V-cycles, GS smoothing iterations, and FGMRES iterations for the 2D Laplace equation. See https://filebin.net/wqv4t3oki732nuq0 for plots.
>
> **GPU Multigrid**
>
> GPU multigrid methods are frequently used in PDE solvers. However, our solver is specialized to work with (possibly non-smooth) neural network outputs over the grid that parameterize the PDEs. The non-smoothness is handled by adding extra smoothing constraints and solving relaxations as saddle-point problems. We are not aware of available solutions for the special case that we consider.
>
> We plan to release the code for our experiments.
>
> **Comments**
>
> By grid size we mean increasing the grid resolution for the same problem and running until convergence. We will make the correction.
>
> The line about derivatives is with reference to the SINDy baselines which estimate numerical derivatives directly on data which makes them susceptible to noise. There is a typo and the word ‘computed’ is missing.
>
> The statement about the parameter $m$ was meant to indicate that it is not a fixed value which implies that fixed basis functions cannot be used to represent the PDE.
>
> We will make the corrections in the next draft.
>
> **Questions**
>
> The V-cycle algorithm itself is fairly standard, however, it is applied in a non-standard setting to solve saddle-point problems in eq 10 and 11. For these problems the standalone V-cycle has subpar performance in our setting. However, we were able to obtain better results using the V-cycle as a preconditioner for FGMRES.
>
> Yes, we also solve the backward pass with the V-cycle (with FGMRES).
>
> We will include more details about the solver implementation in the next draft. The code is in Pytorch together with CuPy for features not available in Pytorch (such as sparse triangular solves). We did not write CUDA kernels, however, that is an option for further improvement. For FGMRES we start with the CPU implementation of GMRES from SciPy. We adapted the GMRES algorithm to FGMRES using the algorithm described in Saad (2003). We ported the code to Pytorch adding batching and GPU support and V-cycle preconditioning, written from scratch.
>
> For simplicity we only threshold once at the end of training (unlike SINDy). The threshold is a hyperparameter per dataset (fixed across noise). We used 0.06, 0.3 and 0.0002 for the easy and hard reaction diffusion, and Navier-Stokes equations. Fine-tuning after thresholding can be used to obtain more accurate coefficients.
>
> The parameterization with a ResNet refers to computing \tilde{u} (line 267 or Figure 1). We parameterize the PDE terms (Figure 1) by a neural network (using \tilde{u}) instead of feeding data (u_data) (which may be noisy) directly. This also helps with training convergence.
>
> Line 411 discusses discovery in the presence of noise. Results are shown in Figure 4 (left). WeakSINDy and MechNN have similar performance with at least TPR 0.6 whereas PDEFIND fails entirely.
>
> We thank the reviewer again.

---

### Decision · Program_Chairs · 2025-05-01

**Decision:**

Accept (poster)

**Comment:**

This paper presents Mechanistic PDE Networks (MechNN-PDE), an extension of the MechNN framework to partial differential equations (PDEs), combining neural representations with a custom GPU-capable sparse multigrid solver. The system enables learning of space-time-dependent linear PDEs from data, allowing for expressive and differentiable modeling of spatiotemporal dynamics, even in noisy regimes. The work targets the discovery (or recovery) of governing PDEs in challenging scenarios, including reaction-diffusion systems and the Navier-Stokes equations.

There was a good discussion around this submission and the authors did a good job in the rebuttal which lead to adjusted scores. The overall consensus is acceptance of the paper and I concur. However, I urge the authors to address the reviewers' point in their revision/camera ready.